# HASH3D: TRAINING-FREE ACCELERATION FOR 3D GENERATION

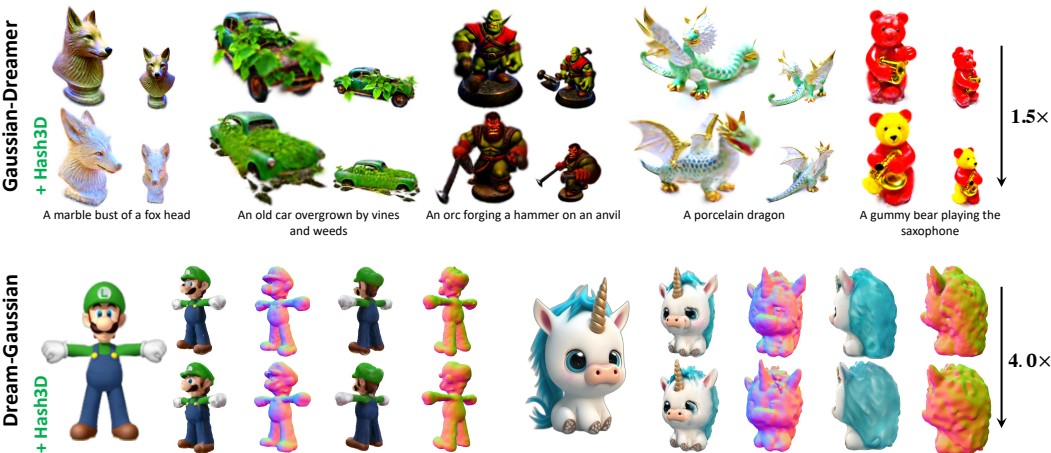

Figure 1: Examples by applying our Hash3D on Gaussian-Dreamer Yi et al. (2023) and Dream-Gaussian Tang et al. (2023). We accelerate Gaussian-Dreamer by $1.5\times$ and Dream-Gaussian by $4\times$ with comparable visual quality.

## ABSTRACT

The quality of 3D generative modeling has been notably improved by the adoption of 2D diffusion models. Despite this progress, the cumbersome optimization process *per se* presents a critical problem to efficiency. In this paper, we introduce Hash3D, a universal acceleration for 3D score distillation sampling (SDS) without model training. Central to Hash3D is the observation that images rendered from similar camera positions and diffusion time-steps often have redundant feature maps. By hashing and reusing these feature maps across nearby timesteps and camera angles, Hash3D eliminates unnecessary calculations. We implement this through an adaptive grid-based hashing. As a result, it largely speeds up the process of 3D generation. Surprisingly, this feature-sharing mechanism not only makes generation faster but also improves the smoothness and view consistency of the synthesized 3D objects. Our experiments covering 5 text-to-3D and 3 image-to-3D models, demonstrate Hash3D's versatility to speed up optimization, enhancing efficiency by $1.5 \sim 4\times$. Additionally, Hash3D's integration with 3D Gaussian splatting largely speeds up 3D model creation, reducing text-to-3D processing to about 10 minutes and image-to-3D conversion to roughly 30 seconds.

## 1 INTRODUCTION

In the evolving landscape of 3D generative modeling, the integration of 2D diffusion models Poole et al. (2023); Wang et al. (2023) has led to notable advancements. These methods leverage off-the-the-shelf image diffusion models to distill 3D models by predicting 2D score functions at different views, known as score distillation sampling (SDS).

While this approach has opened up new opportunities for creating realistic 3D assets, it also brings significant efficiency challenges. Particularly, SDS requires thousands of score predictions from different camera angles and denoising steps in the diffusion model. This results in long optimization times, sometimes taking hours to create a single object Wang et al. (2024). These long durations make them difficult to use in practical applications. We need new solutions to improve its efficiency.

To mitigate this bottleneck, current efforts concentrate on three strategies. The first strategy trains inference-only models Li et al. (2023a); Chen et al. (2023b); Jun & Nichol (2023b); Xu et al. (2024); Liu et al. (2024a) to bypass the lengthy optimization process. While effective, this method requires extensive training time and substantial computational resources. The second approach Tang et al. (2023); Yi et al. (2023); Ren et al. (2023) seeks to reduce optimization times through faster 3D representations. However, each type of representation needs a unique design for 3D generation, which creates its own challenges. The third approach attempts to directly generate sparse views to model 3D objects Kong et al. (2024); Liu et al. (2024b) This method assumes near-perfect consistency for generated views, which, in practice, is often not achievable.

Returning to the core issue within SDS, the major computation is consumed in the repeated sampling of the 2D image score function Song & Ermon (2019). Motivated by methods that accelerate 2D diffusion sampling Song et al. (2021); Bao et al. (2022); Lu et al. (2022), we posed the question: *Is it possible to reduce the number of inference steps of the diffusion model for 3D generation?*

In exploring this question, we make a crucial observation: denoising outputs and feature maps from near camera positions and timesteps are very similar. This discovery led us to develop **Hash3D**, which reduces the computation by leveraging this redundancy.

At its core, Hash3D stores and hashes previously computed features to reduce time. We do this using a a grid-based hash table. Specifically, when a new view is close to one that has already been processed, Hash3D retrieves and reuses the nearby features from the table. This reuse allows Hash3D to compute the current view's score function without repeating earlier calculations.Additionally, we developed a method to dynamically adjust the grid size for each view, which makes the system more adaptable. As a result, Hash3D saves computational resources without requiring any model training or complex changes, making it easy to implement and efficient to use.

Beyond improving efficiency, Hash3D improves the view consistency of generated objects. Traditional diffusion-based methods often result in 3D objects with disjointed appearances when viewed from various angles Armandpour et al. (2023). In contrast, Hash3D links independently generated views by sharing features within each grid. It leads to smoother, more consistent 3D models.

Another key advantage of Hash3D is on its versatility. It integrates seamlessly into a diverse array of diffusion-based 3D generative workflows. Our experiments, covering 5 text-to-3D and 3 image-to-3D models, demonstrate Hash3D's versatility to speed up optimization, enhancing efficiency by $1.3 \sim 4\times$, without compromising on performance. Specifically, the integration of Hash3D with 3D Gaussian Splatting Kerbl et al. (2023) brings a significant leap forward, cutting down the time for text-to-3D to about 10 minutes and image-to-3D to roughly 30 seconds.

The contribution of this paper can be summarized into

- We introduce the Hash3D, a versatile, plug-and-play and training-free acceleration method for diffusion-based text-to-3D and image-to-3D models.
- The paper emphasizes the redundancy in diffusion models when processing nearby views and timesteps. This finding motivates the development of Hash3D, aiming to boost efficiency without compromising quality.
- Hash3D employs an adaptive grid-based hashing to efficiently retrieve features, significantly reducing the computations across view and time.
- Our extensive testing demonstrates that Hash3D not only speeds up the generative process by $1.5 \sim 4\times$, but also results in a slight improvement in performance.

## 2 PRELIMINARY

In this section, we provide the notations and background on optimization-based 3D generation, focusing on diffusion models and Score Distillation Sampling (SDS) Poole et al. (2023).

## 2.1 DIFFUSION MODELS

Diffusion models are generative models that reverse a noise-adding process through a series of latent variables. Starting with data $\mathbf{x}_0 \sim q(\mathbf{x}_0)$, Gaussian noise is progressively added over $T$ steps during the forward process, each defined by $q(\mathbf{x}_t|\mathbf{x}_{t-1}) = \mathcal{N}(\mathbf{x}_t; \sqrt{1-\beta_t}\mathbf{x}_{t-1}, \beta_t\mathbf{I})$, where $\beta_t \in [0,1]$. Due to the Gaussian nature, $\mathbf{x}_t$ can be directly sampled as:

$$\mathbf{x}_t = \sqrt{\bar{\alpha}_t}\mathbf{x}_0 + \sqrt{1-\bar{\alpha}_t}\boldsymbol{\epsilon}, \quad \epsilon \sim \mathcal{N}(0, \mathbf{I}) \tag{1}$$

where $\alpha_t = 1 - \beta_t$ and $\bar{\alpha}_t = \prod_{s=1}^{t} \alpha_s$

The reverse process is modeled as a Markov chain parameterized by a denoising neural network $\boldsymbol{\epsilon}(\mathbf{x}_t, t, y)$, where $y$ is the conditional input, such as text Saharia et al. (2022) or camera pose Liu et al. (2023c). The training of the denoiser aims to minimize a re-weighted evidence lower bound (ELBO), aligning with the noise:

$$\mathcal{L}_{\text{DDPM}} = \mathbb{E}_{t,\mathbf{x}_0,\boldsymbol{\epsilon}}\left[||\boldsymbol{\epsilon} - \boldsymbol{\epsilon}(\mathbf{x}_t, t, y)||_2^2\right] \tag{2}$$

Here, $\boldsymbol{\epsilon}(\mathbf{x}_t, t, y)$ approximates the score function $\nabla_{\mathbf{x}_t} \log p(\mathbf{x}_t|\mathbf{x}_0)$. Data generation is achieved by denoising from noise, often enhanced using classifier-free guidance with scale parameter $\omega$: $\hat{\epsilon}(\mathbf{x}_t, t, y) = (1+\omega)\boldsymbol{\epsilon}(\mathbf{x}_t, t, y) - \omega\boldsymbol{\epsilon}(\mathbf{x}_t, t, \emptyset)$.

**Extracting Feature from Diffusion Model.** A diffusion denoiser $\boldsymbol{\epsilon}$ is typically parameterized with a U-Net Ronneberger et al. (2015). It uses $l$ down-sampling layers $\{D_i\}_{i=1}^{l}$ and up-sampling layers $\{U_i\}_{i=1}^{l}$, coupled with skip connections that link features from $D_i$ to $U_i$. This module effectively merges high-level features from $U_{i+1}$ with low-level features from $D_i$, as expressed by the equation:

$$\mathbf{v}_{i+1}^{(U)} = \text{concat}(D_i(\mathbf{v}_{i-1}^{(D)}), U_{i+1}(\mathbf{v}_i^{(U)})) \tag{3}$$

In this context, $\mathbf{v}_i^{(U)}$ and $\mathbf{v}_{i+1}^{(D)}$ represent the up-sampled and down-sampled features after the $i$-th layer, respectively.

## 2.2 SCORE DISTILLATION SAMPLING (SDS)

The Score Distillation Sampling (SDS) Poole et al. (2023) represents an optimization-based 3D generation method. This method focuses on optimizing the 3D representation, denoted as $\Theta$, using a pre-trained 2D diffusion models with its noise prediction network, denoted as $\boldsymbol{\epsilon}_{\text{pretrain}}(x_t, t, y)$.

Given a camera pose $\boldsymbol{c} = (\theta, \phi, \rho) \in \mathbb{R}^3$ defined by elevation $\phi$, azimuth $\theta$ and camera distances $\rho$, and the its corresponding prompt $y^c$, a differentiable rendering function $g(\cdot; \Theta)$, SDS aims to refine the parameter $\Theta$, such that each rendered image $\boldsymbol{x}_0 = g(\boldsymbol{c}; \theta)$ is perceived as realistic by $\boldsymbol{\epsilon}_{\text{pretrain}}$. The optimization objective is formulated as follows:

$$\min_{\Theta} \mathcal{L}_{\text{SDS}} = \mathbb{E}_{t,\boldsymbol{c}}\left[\frac{\sigma_t}{\alpha_t}\omega(t)\text{KL}\left(q^{\Theta}(\boldsymbol{x}_t|y_c, t) \,\|\, p(\boldsymbol{x}_t|y_c; t)\right)\right] \tag{4}$$

By excluding the Jacobian term of the U-Net, the gradient of the optimization problem can be effectively approximated:

$$\nabla_{\Theta}\mathcal{L}_{\text{SDS}} \approx \mathbb{E}_{t,\boldsymbol{c},\boldsymbol{\epsilon}}\left[\omega(t)(\boldsymbol{\epsilon}_{\text{pretrain}}(\boldsymbol{x}_t, t, y^c) - \boldsymbol{\epsilon})\frac{\partial \boldsymbol{x}}{\partial \Theta}\right] \tag{5}$$

To optimize Eq. 5, we randomly sample different time-step $t$, camera $\boldsymbol{c}$, and random noise $\boldsymbol{\epsilon}$, and compute gradient of the 3D representation, and update $\theta$ accordingly. This approach ensures that the rendered image from 3D object aligns with the distribution learned by the diffusion model.

**Efficiency Problem.** The main challenge lies in the need for thousands to tens of thousands of iterations to optimize Eq 5, each requiring a separate diffusion model inference. This process is time-consuming due to the model's complexity. We make it faster by using a hash function to reuse features from similar inputs, cutting down on the number of calculations needed.

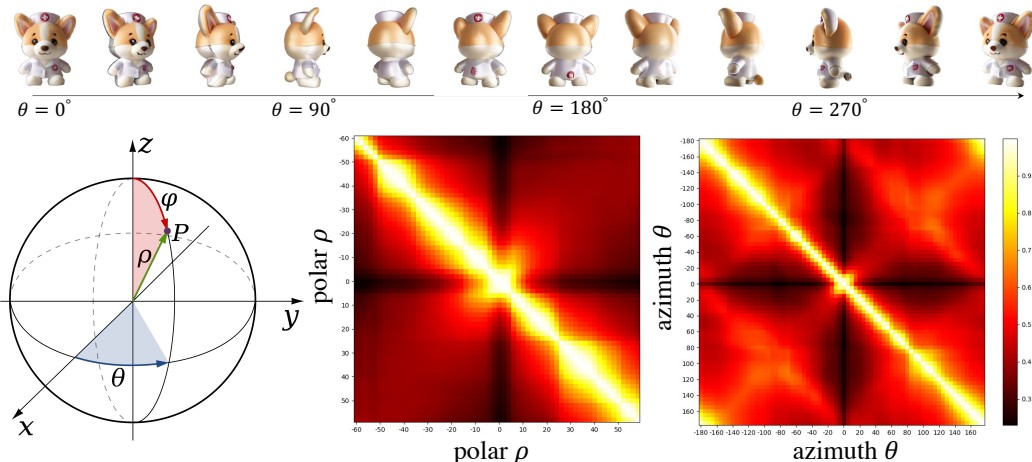

Figure 2: Feature similarity extracted from different camera poses.

# 3 HASH3D

This section introduces Hash3D, , a plug-and-play tool that enhances the efficiency of SDS. We start by analyzing the redundancy presented in the diffusion model when performing 3D generation. Based on the finding, we present our strategy that employs a grid-based hashing to reuse feature across different sampling iterations.

## 3.1 PROBING THE REDUNDANCY IN SDS

Typically, SDS randomly samples camera poses and timesteps to ensure that the rendered views align with the diffusion model's distribution. However, during this repeated sampling, we observe that deep feature extraction at proximate $c$ and $t$ often reveals a high degree of similarity. Therefore, this similarity underpins our method, suggesting that reusing features from nearby points does not significantly impact the model's predictions.

**Measuring the Similarity.** Intuitively, images captured from similar camera positions and at similar times result in similar visual content. We hypothesize that features produced by diffusion models exhibit a similar pattern. Specifically, we propose two hypotheses: (1) *temporal similarity*: features extracted at close timesteps are similar, and (2) *spatial similarity*: features extracted at close estimated camera poses are similar.

Regarding the *temporal similarity*, previous studies Ma et al. (2023); Li et al. (2023b) have noted that features extracted from adjacent timesteps in diffusion models show a high level of similarity.

To test the hypothesis about *spatial similarity*, we conducted a preliminary study using the diffusion model to generate novel views of the same object from different camera positions. Specifically, we used Zero-123 Liu et al. (2023c), which generates images from different camera poses conditioned on a single input image. For each specific camera angle and timestep, we extracted the features $\mathbf{v}_{l-1}^{(U)}$ from the input of the last up-sampling layer. By adjusting elevation angles ($\phi$) and azimuth angles ($\theta$), we were able to measure the cosine similarity of these features between different views, averaging the results across all timesteps.

The findings, presented in Figure 2, reveal a large similarity score in features from views within a $[-10°, 10°]$ range, with the value higher than 0.8. This phenomenon was not unique to Zero-123; we observed similar patterns in text-to-image diffusion models like Stable Diffusion Rombach et al. (2022). These findings underscore the redundancy in predicted outputs within the SDS process.

**Synthesising Novel View for Free.** To leverage redundancy in SDS, we conducted an experiment to create new views by reusing and interpolating scores from precomputed nearby cameras. Specifically, we generated two images using Zero-123 at angles $(\theta, \phi) = (10° \pm \delta, 90°)$ and saved all denoising predictions. By averaging these predictions, we synthesized a third view at $(10°, 90°)$ without additional computation. We experimented with varying $\delta \in \{1°, 5°, 10°, 20°\}$, and compared them with the full denoising predictions.

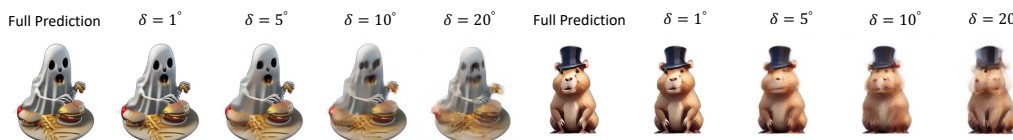

Figure 3: By interpolating latent between generated views, we enable the synthesis of novel views with no computations.

Figure 3 demonstrates that for angles ($\delta$) up to $5°$, novel views closely match fully generated ones, proving effective for closely positioned cameras. Yet, interpolations between cameras at wider angles yield blurrier images. Additionally, optimal window sizes vary by object; for example, a $\delta = 5°$ suits the `ghost` but not the `capybara`, indicating that best window size is sample-specific.

Based on these insights, we presents a novel approach: instead of computing the noise prediction for every new camera pose and timestep, we create a memory system to store previously computed features. As such, we can retrieve and reuse these pre-computed features whenever needed. Ideally, this approach could reduces redundant calculations and speeds up the optimization process.

## 3.2 HASHING-BASED FEATURE REUSE

Based on our analysis, we developed Hash3D, which uses hashing techniques to optimize SDS. Hash3D reduces the repetitive computational cost in diffusion models by trading storage space for faster 3D optimization.

At its core, Hash3D employs a hash table to store and retrieve previously computed features. When Hash3D samples a specific camera pose $c$ and timestep $t$, it first checks the hash table for similar features. If a match is found, it's reused directly in the diffusion model, significantly cutting down on computation. If not, it performs standard inference and adds the new features to the hash table for future use.

**Grid-based Hashing.** To efficiently index the hash table, we use a *grid-based hashing function* based on camera poses $c = (\theta, \phi, \rho)$ and timestep $t$. This function assigns each camera and timestep to a grid cell for data organization and retrieval.

Firstly, we define the size of our grid cells in both the spatial and temporal domains, denoted as $\Delta\theta, \Delta\phi, \Delta\rho$ and $\Delta t$ respectively. For each input key $[\theta, \phi, \rho, t]$, we calculate the grid cell indices:

$$i = \left\lfloor \frac{\theta}{\Delta\theta} \right\rfloor, j = \left\lfloor \frac{\phi}{\Delta\phi} \right\rfloor, k = \left\lfloor \frac{\rho}{\Delta\rho} \right\rfloor, l = \left\lfloor \frac{t}{\Delta t} \right\rfloor \tag{6}$$

These indices are combined into a single hash code: $\texttt{idx} = (i + N_1 \cdot j + N_2 \cdot k + N_3 \cdot l) \mod n$ is used, where $N_1, N_2, N_3$ are large prime numbers Teschner et al. (2003); Nießner et al. (2013), and $n$ denotes the size of the hash table. This hash function maps keys with similar camera poses and timesteps to the same bucket. This grid-based approach not only speeds up data retrieval but also preserves the spatial-temporal relationships in the data, which is crucial for our method.

**Collision Resolution**. When multiple keys are assigned to the same hash value, a collision occurs. We address these collisions using *separate chaining*. In this context, each hash value $\texttt{idx}$ is linked to a distinct queue, denoted as $q_{\texttt{idx}}$. To ensure the queue reflects the most recent data and remains manageable in size, it is limited to a maximum length $Q = 3$. When this limit is reached, the oldest elements is removed to accommodate the new entry, ensuring the queue stays relevant to the evolving 3D representation.

**Feature Retrieval and Update**. After computing the hash value $\texttt{idx}$, we either retrieve features from the hash table or update it with new ones. We control this with hash probability $0 < \eta < 1$. With probability $\eta$, we retrieve features; otherwise, we perform an update.

For feature updates, following prior work Ma et al. (2023), we extract the feature $\mathbf{v}_{l-1}^{(U)}$, which is the input of the last up-sampling layer in the U-net. Once extracted, we compute the hash code $\texttt{idx}$ and append the data to the corresponding queue $q_{\texttt{idx}}$. The stored data includes noisy latent input $x$, camera pose $c$, timestep $t$, and extracted diffusion features $\mathbf{v}_{l-1}^{(U)}$.

Figure 4: Overall pipeline of our Hash3D. Given the sampled camera and time-step, we retrieve the intermediate diffusion feature from hash table. If no matching found, it performs a standard inference and stores the new feature in the hash table; otherwise, if a feature from a close-up view already exists, it is reused without re-calculation.

For feature retrieval, we aggregate data from $q_{\text{idx}}$ through weighted averaging. This method considers the distance of each noisy input $\boldsymbol{x}_i$ from the current query point $\boldsymbol{x}$. The weighted average $\mathbf{v}$ for a given index is calculated as follows:

$$\mathbf{v} = \sum_{i=1}^{|q_{\text{idx}}|} W_i \mathbf{v}_i, \text{ where } W_i = \frac{e^{(-||\boldsymbol{x}-\boldsymbol{x}_i||_2^2)}}{\sum_{i=1}^{|q_{\text{idx}}|} e^{(-||\boldsymbol{x}-\boldsymbol{x}_i||_2^2)}} \tag{7}$$

Here, $W_i$ is the weight assigned to $\mathbf{v}_i$ based on its distance from the query point, and $|q_{\text{idx}}|$ is the current length of the queue. An empty queue $|q_{\text{idx}}|$ indicates unsuccessful retrieval, necessitating feature update.

### 3.3 ADAPTIVE GRID HASHING

In grid-based hashing, the selection of an appropriate grid size $\Delta\theta, \Delta\phi, \Delta\rho, \Delta t$ — plays a pivotal role. As illustrated in Section 3.1, we see three insights related to grid size. First, feature similarity is only maintained at a median grid size; overly large grids tend to produce artifacts in generated views. Second, it is suggested that ideal grid size differs across various objects. Third, even for a single object, optimal grid sizes vary for different views and time steps, indicating the necessity for adaptive grid sizing to ensure optimal hashing performance.

**Learning to Adjust the Grid Size.** To address these challenges, we propose to dynamically adjusting grid sizes. The objective is to maximize the average cosine similarity $\cos(\cdot, \cdot)$ among features within each grid. In other words, only if the feature is similar enough, we can reuse it. Such problem is formulated as

$$\max_{\Delta\theta,\Delta\phi,\Delta\rho,\Delta t} \frac{1}{|q_{\text{idx}}|} \sum_{i,j}^{|q_{\text{idx}}|} \cos(\mathbf{v}_j, \mathbf{v}_i), \quad s.t.|q_{\text{idx}}| > 0 \quad \text{[Non-empty]} \tag{8}$$

Given our hashing function is *non-differentiale*, we employ a brute-force approach. Namely, we evaluate $M$ predetermined potential grid sizes, each corresponding to a distinct hash table, and only use best one.

For each input $[\theta, \phi, \rho, t]$, we calculate the hash code $\{\text{idx}^{(m)}\}_{m=1}^{M}$ for $M$ times, and indexing in each bucket. Feature vectors are updated accordingly, with new elements being appended to their respective bucket. We calculate the cosine similarity between the new and existing elements in the bucket, maintaining a running average $s_{\text{idx}^{(n)}}$ of these similarities

$$s_{\text{idx}^{(m)}} \leftarrow \gamma s_{\text{idx}^{(m)}} + (1-\gamma)\frac{1}{|q_{\text{idx}^{(m)}}|} \sum_{i=1}^{|q_{\text{idx}^{(m)}}|} \cos(\mathbf{v}_{new}, \mathbf{v}_i) \tag{9}$$

During retrieval, we hash across all $M$ grid sizes but only consider the grid with the highest average similarity for feature extraction.

**Computational and Memory Efficiency.** Despite employing a brute-force approach that involves hashing $M$ times for each input, our method maintains computational efficiency due to the low cost of hashing. It also maintains memory efficiency, as hash tables store only references to data. To prioritize speed, we deliberately avoid using neural networks for hashing function learning.

## 4 EXPERIMENT

In this section, we assess the effectiveness of our HS by integrating it with various 3D generative models, encompassing both image-to-3D and text-to-3D tasks.

### 4.1 EXPERIMENTAL SETUP

**Baselines.** To verify our method, we conduct extensive tests across a wide range of baseline text-to-3D and image-to-3D methods.

- **Image-to-3D.** We build our method on Zero-123+SDS Liu et al. (2023b), DreamGaussian Tang et al. (2023) and Magic123 Qian et al. (2024). For Zero-123+SDS, we incorporate Instant-NGP Müller et al. (2022) and Gaussian Splatting Kerbl et al. (2023) as its representation. We call these two variants Zero-123 (NeRF) and Zero-123 (GS).

- **Text-to-3D.** Our tests also covered a range of methods, such as Dreamfusion Poole et al. (2023), Fantasia3D Chen et al. (2023a), Latent-NeRF Metzer et al. (2023), Magic3D Lin et al. (2023), and GaussianDreamer Yi et al. (2023).

For DreamGaussian and GaussianDreamer, we implement Hash3D on top of the official code. And for other methods, we use the reproduction from `threestudio`[1].

**Implementation Details.** We stick to the same hyper-parameter setup within their original implementations of these methods. For text-to-3D, we use the `stable-diffusion-2-1`[2] as our 2D diffusion model. For image-to-3D, we employ the `stable-zero123`[3]. We use a default hash probability setting of $\eta = 0.1$. We use $M = 3$ sets of grid sizes, with $\Delta\theta, \Delta\phi, \Delta t \in \{10, 20, 30\}$ and $\Delta\rho \in \{0.1, 0.15, 0.2\}$. We verify this hyper-parameter setup in the ablation study.

**Dataset and Evaluation Metrics.** To assess our method, we focus on evaluating the computational cost and visual quality achieved by implementing Hash3D.

- **Image-to-3D.** For image-to-3D experiments, we used the Google Scanned Objects (GSO) dataset Downs et al. (2022) for evaluation Liu et al. (2024a; 2023c). We evaluated novel view synthesis (NVS) performance with PSNR, SSIM Wang et al. (2004), and LPIPS Zhang et al. (2018). We selected 30 objects, each with a $256^2$ input image for 3D reconstruction. We rendered 16 views at a 30-degree elevation with varying azimuths to compare the reconstructions with ground truth. CLIP-similarity scores were calculated to ensure semantic consistency between rendered views and original images.

- **Text-to-3D.** We generated 3D models from 50 different prompts, selected based on a prior study. To evaluate our methods, we focused on two primary metrics: mean±std CLIP-similarity Radford et al. (2021); Qian et al. (2023); Liu et al. (2023a) and the average generation time for each method. CLIP-similarity was measured between the input prompt and 8 uniformly rendered views.

- **User Study.** To evaluate the visual quality of generated 3D objects, we conducted a study with 44 participants. They viewed 12 video renderings from two methods: Zero-123 (NeRF) for images-to-3D and Gaussian-Dreamer for text-to-3D, with and without Hash3D. Participants rated each pair by distributing 100 points to indicate perceived quality differences.

- **Computational Cost.** We report the running time for each experiment on a single RTX A5000 and include MACs in the tables. As feature retrieval is stochastic, we provide the theoretical average MACs, assuming all retrievals succeed.

### 4.2 3D GENERATION RESULTS

**Image-to-3D Qualitative Results.** Figure 5 shows the results of integrating Hash3D into the Zero-123 framework for generating 3D objects. This integration maintains visual quality and view consistency while significantly reducing processing time. In some cases, Hash3D outperforms the baseline, such as the clearer "dragon wing boundaries" in row 1 and the more distinct "train taillights"

---

[1]https://github.com/threestudio-project/threestudio

[2]https://huggingface.co/stabilityai/stable-diffusion-2-1

[3]https://huggingface.co/stabilityai/stable-zero123

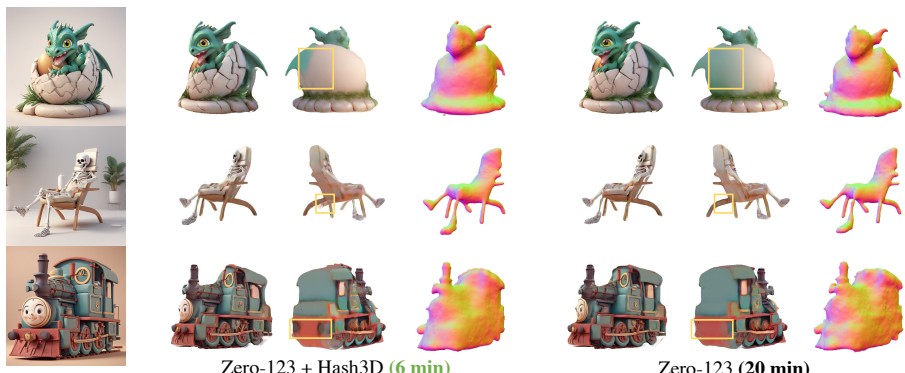

Zero-123 + Hash3D **(6 min)**          Zero-123 **(20 min)**

Figure 5: Qualitative Results using Hash3D along with Zero123 for image-to-3D generation. We mark the visual dissimilarity in yellow.

Table 1: Speed and performance comparison when integrated image-to-3D models with Hash3D. We report the original running time in their paper.

| Method | Time↓ | Speed↑ | MACs↓ | PSNR↑ | SSIM↑ | LPIPS↓ | CLIP-G/14↑ |
|---|---|---|---|---|---|---|---|
| DreamGaussian | 2m | - | 168.78G | 16.202±2.501 | 0.772±0.102 | 0.225±0.111 | 0.693±0.105 |
| + **Hash3D** | **30s** | **4.0×** | 154.76G | **16.356**±2.533 | **0.776**±0.103 | **0.223**±0.113 | **0.694**±0.104 |
| Zero-123(NeRF) | 20m | - | 168.78G | 17.773±3.074 | 0.787±0.101 | 0.198±0.097 | 0.662±0.0107 |
| + **Hash3D** | **7m** | **3.3×** | 154.76G | **17.961**±3.034 | **0.789**±0.095 | **0.196**±0.0971 | **0.665**±0.104 |
| Zero-123(GS) | 6m | - | 168.78G | 18.409±2.615 | 0.789±0.100 | **0.204**±0.101 | **0.643**±0.105 |
| + **Hash3D** | **3m** | **2.0×** | 154.76G | **18.616**±2.898 | **0.793**±0.099 | **0.204**±0.099 | 0.632±0.106 |
| Magic123 | 120m | - | 847.38G | **18.718**±2.446 | **0.803**±0.093 | **0.169**±0.092 | **0.718**±0.099 |
| + **Hash3D** | **74m** | **1.6×** | 776.97G | 18.631±2.726 | **0.803**±0.091 | 0.174±0.093 | 0.715±0.107 |

in row 4. Similar visual fidelity is seen in Figure 1, where Hash3D is used with DreamGaussian, demonstrating effective quality maintenance and improved efficiency.

**Image-to-3D Quantitative Results.** Table 1 presents a detailed numerical analysis of novel view synthesis, including CLIP scores and running times for all four baseline methods. Notably, Our method achieves a $4\times$ speedup on DreamGaussian and $3\times$ on Zero-123 (NeRF), due to Hash3D's efficient feature retrieval and reuse. This not only accelerates processing but also slightly improves CLIP score performance by sharing features across views, reducing inconsistencies, and producing smoother 3D models.

**Text-to-3D Qualitative Results.** In Figure 6, we present the results generated by our Hash3D, on top of DreamFusion Poole et al. (2023), SDS+GS, and Fantasia3D Chen et al. (2023a). It demonstrates that Hash3D maintains comparable visual quality to these established methods.

**Text-to-3D Quantitative Results.** Table 2 presents a quantitative evaluation of Hash3D. Hash3D significantly reduces processing times across various methods while maintaining visual quality, with minimal impact on CLIP scores. For methods like GaussianDreamer, it even slightly improves visual fidelity, indicating the benefit of leveraging relationships between nearby camera views.

**User preference study.** As shown in Figure 7, Hash3D received an average preference score of 52.33/100 and 56.29/100 when compared to Zero-123 (NeRF) and Gaussian-Dreamer. These scores are consistent with previous results, indicating that Hash3D slightly enhances the visual quality of the generated objects.

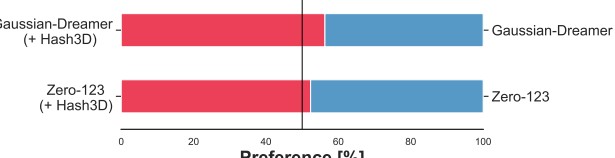

Figure 7: User preference study for Hash3D.

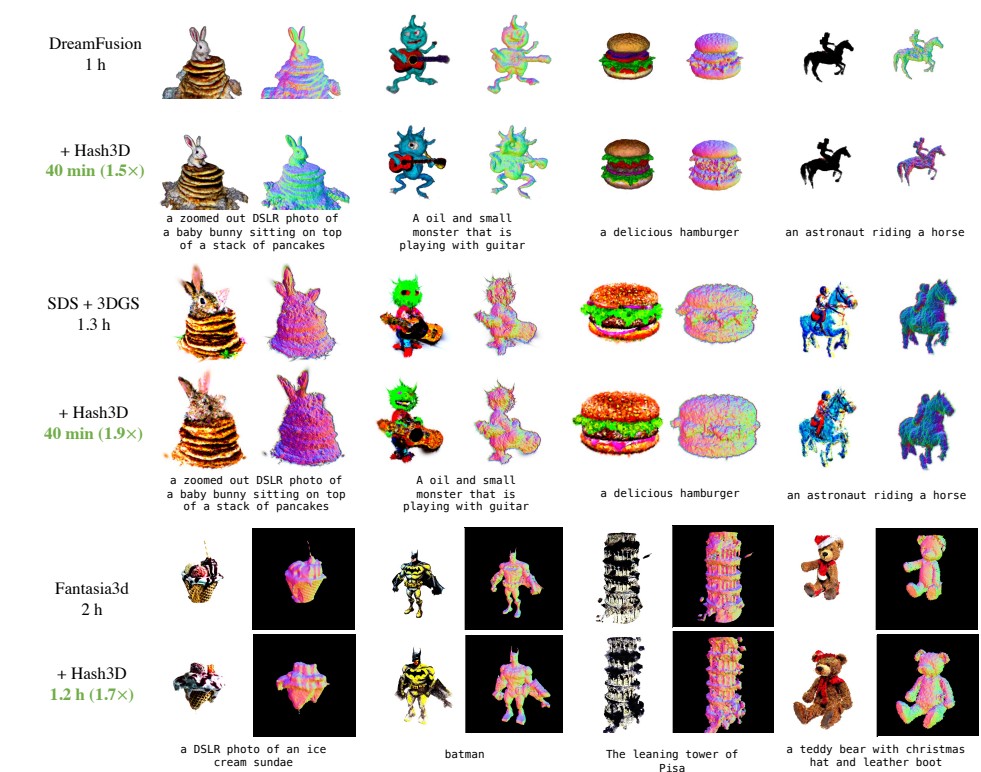

Figure 6: Visual comparison for text-to-3D task, when applying Hash3D to DreamFusion Poole et al. (2023), SDS+GS and Fantasia3D Chen et al. (2023a).

Table 2: Speed and performance comparison between various text-to-3D baseline when integrated with Hash3D.

| Method | Time↓ | Speed↑ | MACs↓ | CLIP-G/14↑ | CLIP-L/14↑ | CLIP-B/32↑ |
|---|---|---|---|---|---|---|
| Dreamfusion | 1h 00m | - | 678.60G | 0.407± 0.088 | **0.267**±0.058 | **0.314** ±0.049 |
| **+ Hash3D** | **40m** | **1.5×** | 622.21G | **0.411**±0.070 | 0.266± 0.050 | 0.312±0.044 |
| Latent-NeRF | 30m | - | 678.60G | **0.406**±0.033 | 0.254±0.039 | 0.306±0.037 |
| **+ Hash3D** | **17m** | **1.8×** | 622.21G | **0.406**±0.038 | **0.258**±0.045 | **0.305**±0.038 |
| SDS+GS | 1h 18m | - | 678.60G | **0.413**±0.048 | **0.263**±0.034 | **0.313**±0.036 |
| **+ Hash3D** | **40m** | **1.9×** | 622.21G | 0.402±0.062 | 0.252±0.041 | 0.306±0.036 |
| Magic3D | 1h 30m | - | 678.60G | **0.399**±0.012 | **0.257**±0.064 | **0.303**±0.059 |
| **+ Hash3D** | **1h** | **1.5×** | 622.21G | 0.393±0.011 | 0.250±0.054 | 0.304±0.052 |
| GaussianDreamer | 15m | - | 678.60G | 0.412±0.049 | 0.267±0.035 | 0.312±0.038 |
| **+ Hash3D** | **10m** | **1.5×** | 622.21G | **0.416**±0.057 | **0.271**±0.036 | 0.312±0.037 |

## 4.3 ABLATION STUDY AND ANALYSIS

In this section, we study several key components in our Hash3D framework.

**Ablation 1: Hashing *vs.* Storing All Features.** We compare hashing features with storing all past features and retrieving them by similarity. As shown in Table 3, hashing is more effective. On efficiency side, storing all feature even causes an OOM error in Dreamfusion. Hashing requires only constant space. Additionally, our grid-based hashing leverages geometric information to improve sample quality. More visual results are available in the appendix.

Table 3: Comparison of feature retrieval with and without hashing.

| Name | Time↓ | GPU Mem.↓ | CLIP-G/14↑ |
|---|---|---|---|
| Hash3D+Zero-123 (NeRF) w/o hashing | 11m | 8G | 0.661±0.096 |
| Hash3D+Zero-123 (NeRF) | **7m** | **6G** | **0.665**±0.104 |
| Hash3D+DreamFusion w/o hashing | - | OOM | - |
| Hash3D+DreamFusion | **40m** | **8G** | **0.411**±0.070 |

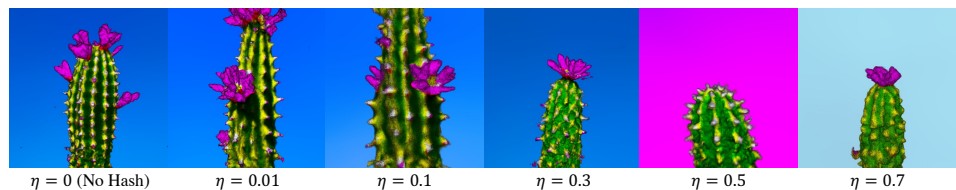

| $\eta = 0$ (No Hash) | $\eta = 0.01$ | $\eta = 0.1$ | $\eta = 0.3$ | $\eta = 0.5$ | $\eta = 0.7$ |

Figure 8: Qualitative comparison with different hash probability $\eta$.

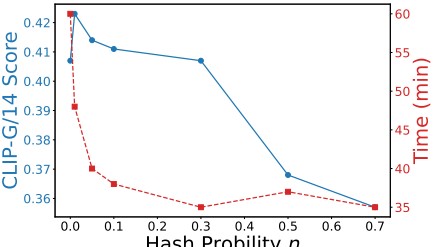

Figure 9: Ablation study with different hash probability $\eta$.

| Method | Time | CLIP-G/14 |
|---|---|---|
| Zero-123 (NeRF) + Hash3D w/n | 6 min | $0.631\pm0.090$ |
| Zero-123 (NeRF) + Hash3D | 7 min | $\mathbf{0.665}\pm0.104$ |
| Zero-123 (GS) + Hash3D w/n | 3 min | $0.622\pm0.083$ |
| Zero-123 (GS) + Hash3D | 3 min | $\mathbf{0.632}\pm1.06$ |

Figure 10: Comparison between Hashing Features *vs.* Hashing Noise, applied to Zero-123.

**Ablation 2: Hashing Features *vs.* Hashing Noise.** In Hash3D, we hash intermediate features within the diffusion U-Net. Alternatively, we developed Hash3D with noise (Hash3D w/n), which hashes and reuses the denoising prediction directly. We tested both methods on the image-to-3D task using Zero123, with results shown in Table 10. Interestingly, while Hash3D w/n reduced processing time, it significantly lowered CLIP scores. This highlights that hashing features is more effective than hashing noise predictions.

**Ablation 3: Influence of Hash Probability $\eta$.** A key parameter in Hash3D is the feature retrieval probability $\eta$. We tested $\eta \in \{0.01, 0.05, 0.1, 0.3, 0.5, 0.7\}$ using Dreamfusion. As shown in Figure 9, runtime decreases as $\eta$ increases. Generated objects are visualized in Figure 8. For $\eta < 0.3$, Hash3D also improved the visual quality of 3D models by enabling smoother noise predictions through feature sharing. However, for $\eta > 0.3$, the runtime gains were minimal. This balance of performance and efficiency led us to choose $\eta = 0.1$ for our main experiments.

**Ablation 4: Adaptive Grid Size.** We introduce AdaptGrid, which dynamically adjusts the grid size for hashing based on each sample. Compared to using a constant grid size in Dreamfusion, AdaptGrid performs better as shown in Table 4. Larger grid sizes reduce the visual quality of 3D objects, while smaller grid sizes maintain quality but increase computation time because fewer features match. AdaptGrid effectively balances visual quality and efficiency by optimizing the grid size for each sample.

Table 4: Ablation study on the Adaptive *v.s.* Constant Grid Size.

| $\Delta\theta, \Delta\phi, \Delta\rho, \Delta t$ | (10, 10, 0.1, 10) | (20, 20, 0.15, 20) | (30, 30, 0.2, 30) | AdaptGrid (Ours) |
|---|---|---|---|---|
| **CLIP-G/14**↑ | $0.408\pm0.033$ | $0.345\pm0.055$ | $0.287\pm0.078$ | $\mathbf{0.411}\pm0.070$ |
| **Time**↓ | 48m | 38m | 32m | 40m |

## 5 CONCLUSION

In this paper, we present Hash3D, a training-free technique that improves the efficiency of diffusion-based 3D generative modeling. Hash3D utilizes adaptive grid-based hashing to efficiently retrieve and reuse features from adjacent camera poses, to minimize redundant computations. As a result, Hash3D not only speeds up 3D model generation by $1.5 \sim 4\times$ without the need for additional training, but it also improves the smoothness and consistency of the generated 3D models.

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

## A  APPENDIX

In this document, we provide additional information and analysis for our proposed Hash3D. We begin by describing how the feature is extraction from diffusion model in Section B. Following that, we delve into further analysis for Hash3D, including ablation studies in Section C, and provide visualizations in Section D. More implementation details are disclosed in Section E, which also includes the pseudo-code for our hash table data structure and the feature hashing process in Section F. For additional information, please refer to the source code available in the uploaded files.

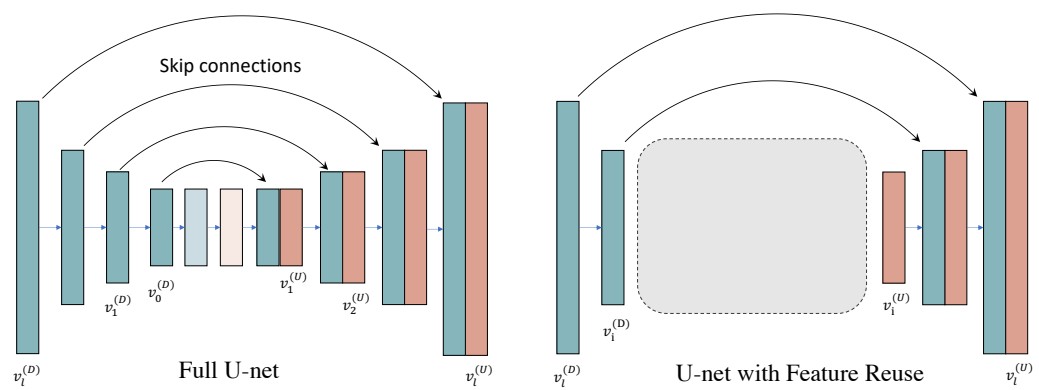

Figure 11: Structure of the U-Net and our feature extraction setup.

## B DETAILS FOR FEATURE EXTRACTION

As Hash3D involves the extraction of features from U-Net, we here introduce how we define and indexing those features. As illustrated in Figure 11, we adopt the definition that, the indices for the downsampling layers are arranged in decreasing order, whereas for the upsampling layers, the indices follow an increasing order. With in total $l$ up-sample layers and $l$ down-sample layers, the skip connection merges high-level features from $U_{i+1}$ with low-level features from $D_i$, as expressed by the equation:

$$\mathbf{v}_{i+1}^{(U)} = \text{concat}(D_i(\mathbf{v}_{i-1}^{(D)}), U_{i+1}(\mathbf{v}_i^{(U)})) \tag{10}$$

If we would like to reuse the feature $\mathbf{v}_i^{(U)}$ from the U-Net, upon retrieval, the model only requires the forwarding of layers $D_l$ to $D_i$ and of $U_{i+1}$ to $U_l$. This approach allows us to bypass all intermediate computational blocks, enhancing efficiency.

## C ANALYSIS AND ABLATION STUDY

### C.1 KEY-BASED HASHING & CONTENT-BASED AGGREGATION

In fact, Hash3D utilizes a hierarchical process for feature reuse, involving a *key-based* hashing stage and a *content-based* feature aggregation stage. In the first stage of key-based hashing, Hash3D computes a hash code corresponding to a bucket according to the camera pose and time step. This efficiently retrieves a set of candidate features. Subsequently, Hash3D performs a content-based refinement within the retrieved bucket. Features are aggregated based on the similarity (distance) between their input latents.

This section investigates the effectiveness of the two-stage hashing.

**Experimental setup.** To assess the contribution of each hashing stage, we conducted two experiments:

- **Ablation 1: Removing Key-based Hashing.** In this experiment, we removed the key-based hashing stage. Instead, the query feature's latent vector was directly compared against the entire pre-extracted feature pool (no hashing at all). To achieve this, we established a queue with maximum length of 1000 to store all previously extracted features.

- **Ablation 2: Removing Content-based Aggregation.** Here, we omitted the content-based aggregation stage. As replacement, within each bucket, only the features with closest hash key (camera pose and timestep) will be returned.

We test it on Zero-123 (NeRF) and compare the visual fidelity.

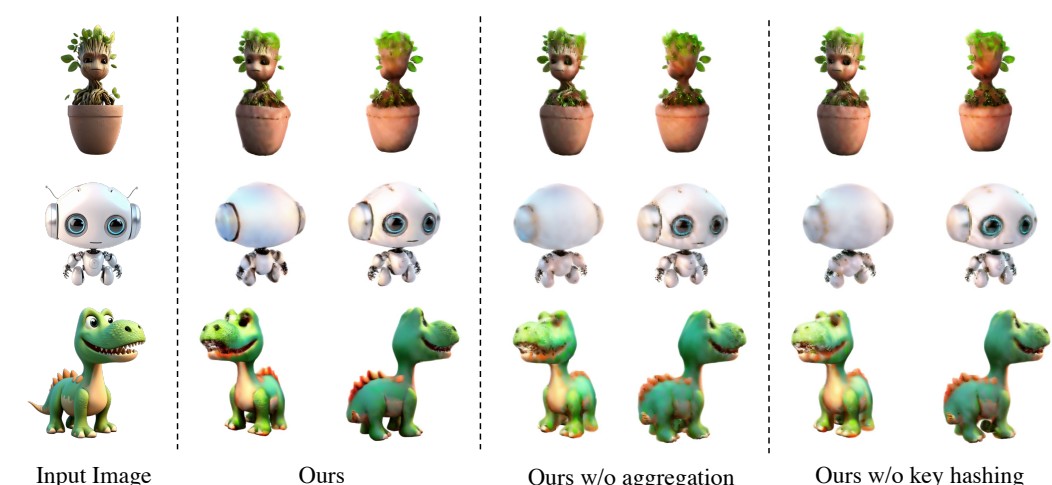

Input Image          Ours          Ours w/o aggregation          Ours w/o key hashing

Figure 12: Results with different hashing strategy. "Our w/o aggregation" is short for "Ours without feature aggregation" and "Ours w/o key hashing" is for "Ours without key hashing".

**Results.** Our study presents visualization for various retrieval strategies, as shown in Figure 12. We refer to our first variation as "Ours without key hashing" and the second as "Ours without feature aggregation".

It is observed that our complete solution achieves the highest visual fidelity. Interestingly, the exclusion of feature aggregation leads to the emergence of moiré patterns, exemplified by the `eye of the robot`. This phenomenon occurs because multiple hash keys can map to the same cached feature, resulting in overlapping patterns in the generated images. On the contrary, the omission of the key-based hashing stage produces images that are overly smooth and lack detail. By first filtering features within a grid and subsequently aggregating them based on latent similarity, our method ensures clearer boundaries of the generated objects.

## C.2 HASHING FEATURE *vs.* HASHING NOISE

Beyond the quantitative results presented in Table 9 of the main paper, we offer visual comparisons between hashing features and hashing denoising predictions in Figure 13. We implement Hash3D on top of Zero-123 (NeRF) and visualize the multiview images of the reconstructed objects.

Hashing noise leads to the generation of saturated 3D objects, occasionally exhibiting mosaic patterns. Although this method proves to be slightly faster, it compromises visual quality, aligning with our quantitative findings. Consequently, we advocate for the use of feature hashing in our study, as it maintains higher fidelity in the visual results.

## C.3 OPTIMAL LAYER FOR FEATURE EXTRACTION

In caching and retrieving features within diffusion models, a critical question arises: *which layer's features should be extracted?* Ideally, extracting features from deeper layers, closer to the output, can significantly reduce computational overhead but might result in a slight loss of fidelity in the predicted images. On the other hand, hashing features from earlier, low-level layers retains higher performance at the cost of increased inference overhead. This presents a trade-off between computational efficiency and output quality. We in this section valid our selection.

For example, the Zero123 U-Net contains 10 skip connections, each associated with a down-sampling layer and a up-sampling layers. We test 10 positions for feature extraction, and show the results.

Figure 14 illustrates that, generally, a larger layer index $i$—indicating proximity to the output—results in reduced optimization time but slightly diminished visual quality. However, given

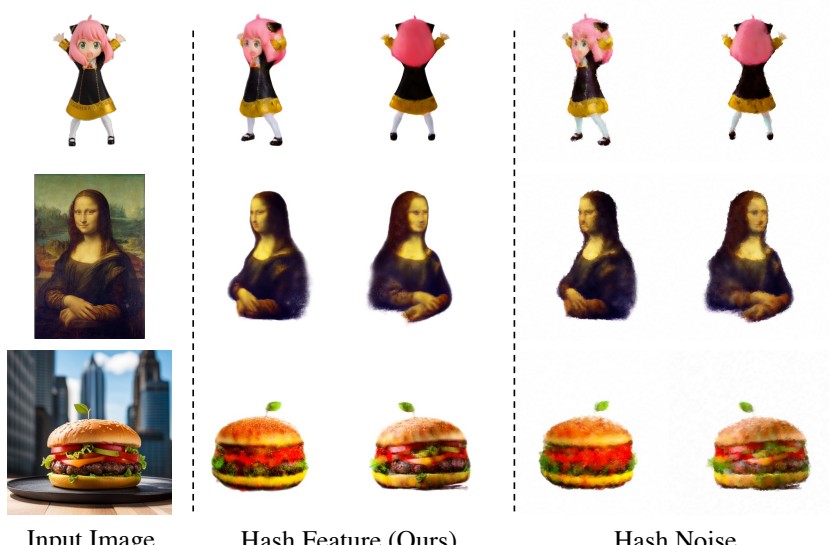

| Input Image | Hash Feature (Ours) | Hash Noise |
| --- | --- | --- |

Figure 13: Results when hashing features or hashing the desnoising predictions.

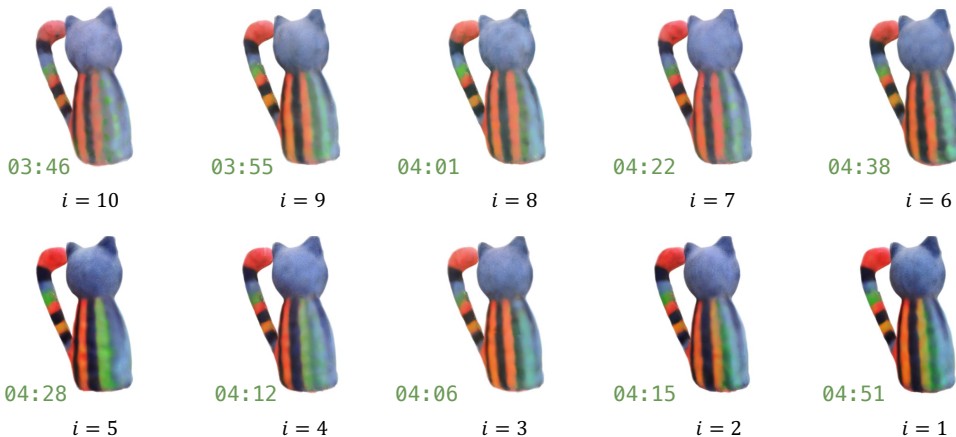

Figure 14: Impact of feature hashing at various layers on optimization time and visual fidelity. Note that, *larger layer index indicating closer to the output, with smaller computation.*

the minimal impact on fidelity, we opt for using $i = 10$, the layer before the last upsampling, for feature extraction in our experiments. This choice effectively balances computational efficiency with the maintenance of high visual quality.

## D ADDITIONAL RESULTS

This section presents further visualizations demonstrating the effectiveness of our method. Specifically, we compare our Hash3D+Zero123 approach with the original Zero-123 method in the context of image-to-3D reconstruction, as illustrated in Figure 15. Additionally, we evaluate our method against Gaussian-Dreamer for text-to-3D generation, as shown in Figure 16. Our results showcase superior visual quality: we achieve this in 7 minutes compared to Zero-123's 20 minutes, and in 10 minutes against Gaussian-Dreamer's 15 minutes.

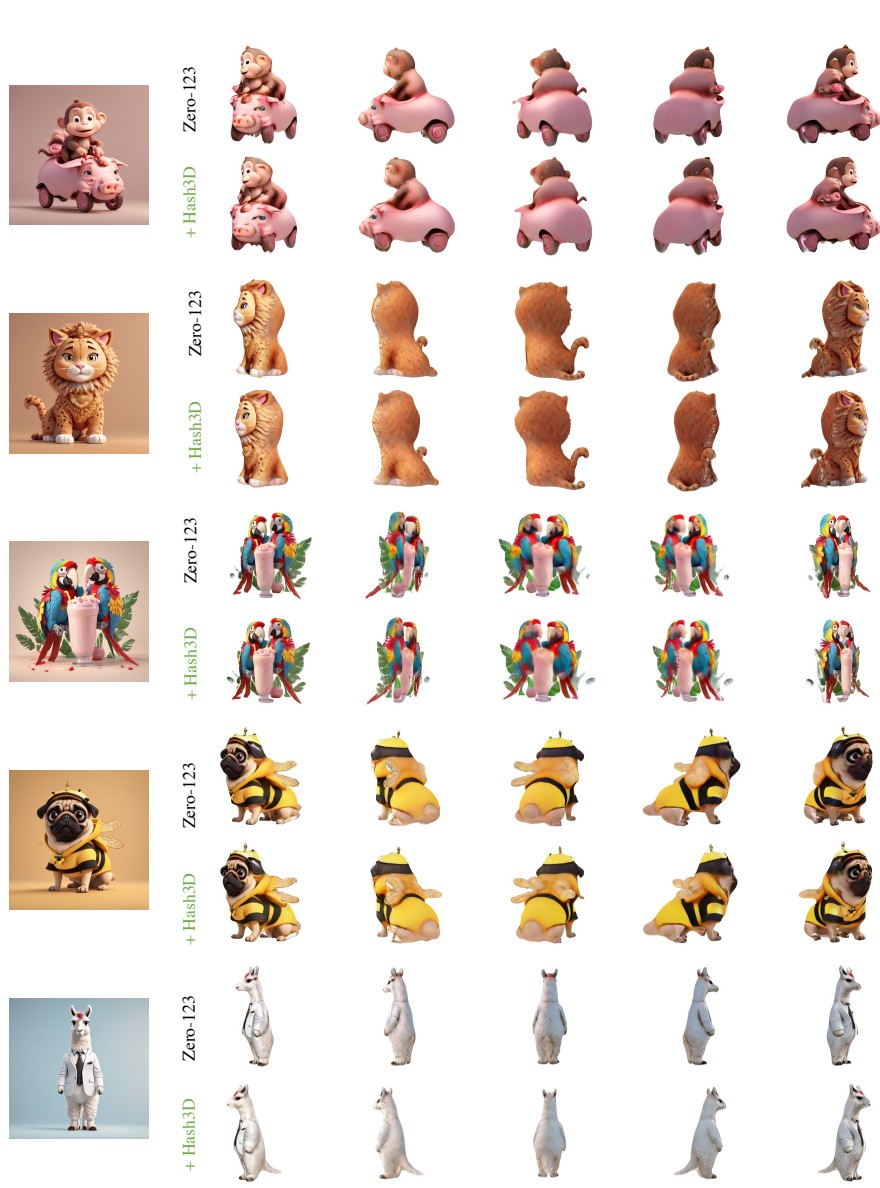

Figure 15: Qualitative Comparison when applying Hash3D on top of Zero-123.

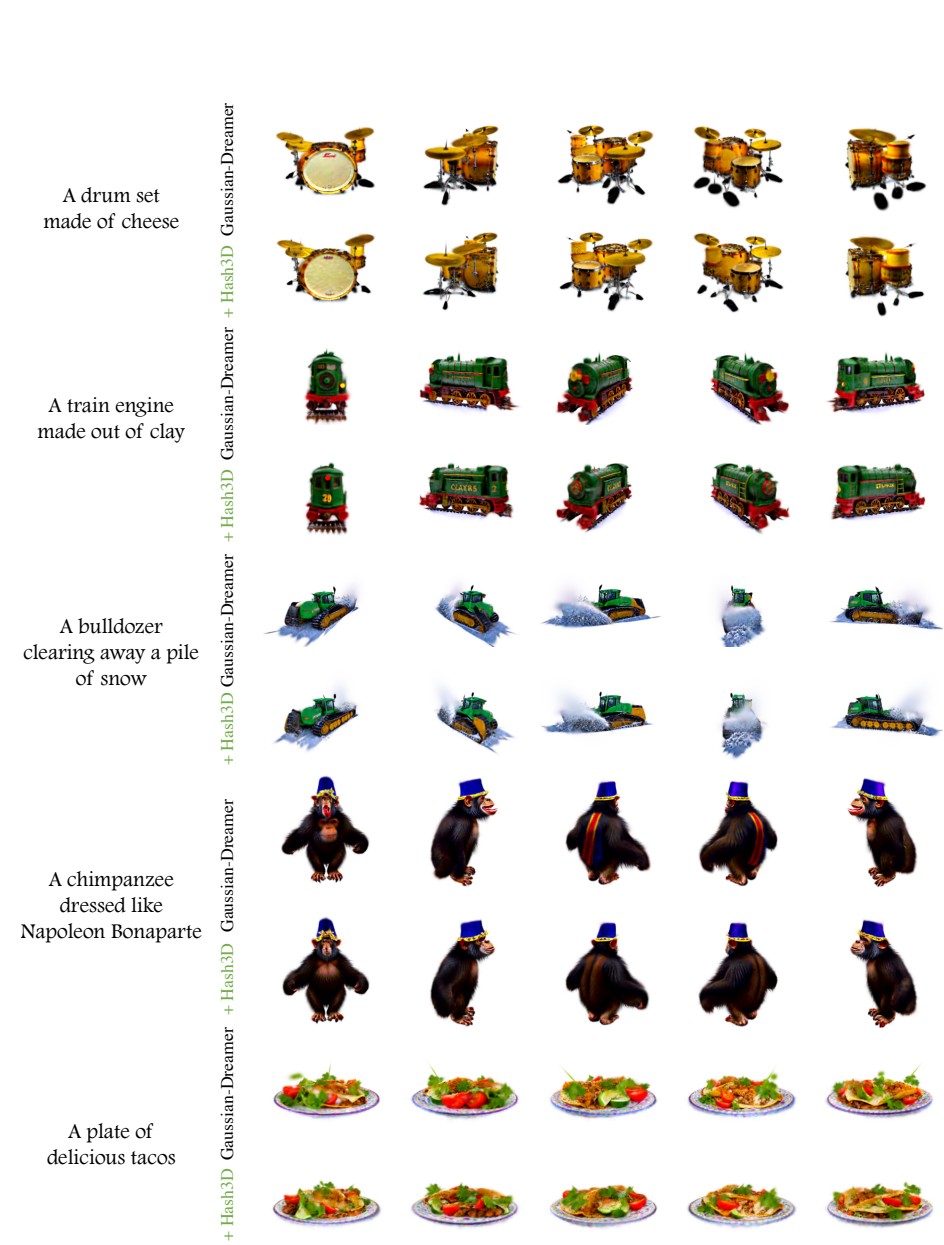

Figure 16: Qualitative Comparison when applying Hash3D on top of Gaussian-Dreamer.

## E   IMPLEMENTATION DETAILS

We use the official implementation for Dream-Gaussian and Gaussian-Dreamer. For all other methods, we take the `threestudio`'s implementations, with their default experimental configurations.

**Image-to-3D**:

- **Zero-123 (NeRF)**: We employ NeRF with hash grid encoding for the 3D representation. We leverage `stable-zero123` as the diffusion model to optimize this representation using the SDS loss. A classifier-free guidance of 3.0 is used, and the Adam optimizer updates the parameters for 1,000 steps with learning rate of 0.01. We use a batch size of 1.

- **Zero-123 (GS)**: We employ Gaussian Splatting for the 3D representation. For other details, we follow the setup for Zero-123 (NeRF). We use the implantation from `threestudio-3dgs` [4].

- **Dream-Gaussian**: We use the official implementation [5]. The initial Gaussians consists of 5,000 randomly colored points on a sphere. In the first stage, we update the parameters for 500 iterations using `stable-zero123` model and the SDS loss. The second stage focuses on refining the mesh for 50 additional steps with the RGB MSE loss. Since this stage doesn't require the SDS loss, we employ Deepcache Ma et al. (2023) for acceleration. Deepcache can be considered a simplified version of our Hash3D, focusing solely on temporal reuse.

- **Magic-123**: Following the configurations from `threestudio`, we use `stable-diffusion-v1-5` as the text-to-image diffusion model, and `stable-zero123` as the image-to-3D diffusion model. In the first stage, both models work together to optimize a NeRF as the 3D representation for 10,000 iterations. This NeRF is then converted into an explicit surface mesh representation Shen et al. (2021) in the second stage, which also undergoes optimization for another 10,000 iterations. Both stages use the SDS loss, where the loss weights for text-to-image and image-to-3D diffusion are set to 0.025 and 0.1.

**Text-to-3D**:

- **Dreamfusion**: We use the `stable-diffusion-2-1-base` to optimize the NeRF representation with hash encoding, using SDS loss. We apply a classifier-free guidance technique, setting its scale to 100. For the optimization process, we use the Adam optimizer with a learning rate of 0.01 and run the process for a total of 10,000 iterations.

- **Latent-NeRF**: We use the same setup as in above Dreamfusion experiment, except that we use a vallina NeRF representation.

- **SDS+GS**: Compared to the Dreamfusion above, the only difference is that we use a 3D Gaussian Splatting to represent the 3D object. The 3D Gaussians are initialized from the shap-e Jun & Nichol (2023a) predicted mesh. We use the implementation from `threestudio-3dgs`.

- **Magic3D**: The first stage of Magic3D involves updating an instant-npg like NeRF representation for 10,000 iterations, using the `stable-diffusion-2-1-base` model and SDS loss. Subsequently, this NeRF is converted into an explicit surface mesh, which is then optimized for an additional 10,000 iterations.

- **GaussianDreamer**: We take the official implementation [6] to do the experiments. The Gaussian points are initialized from shap-e Jun & Nichol (2023a) predicted mesh. Optimization is conducted over 1,200 steps using the `stable-diffusion-2-1-base` model with a classifier-free guidance scale of 100, and Adam optimization at a learning rate of 0.001.

---

[4]https://github.com/DSaurus/threestudio-3dgs

[5]https://github.com/dreamgaussian/dreamgaussian/tree/main

[6]https://github.com/hustvl/GaussianDreamer

## F PSEUDO-CODE FOR HASH3D

In our paper, we introduce a core mechanism that utilizes a grid-based hashing table to organize features extracted across various camera poses and time steps. This section provides a detailed overview, including pseudo-code, for two main components: (1) the data structure and associated functions of our grid-based hashing, in Listing 1, and (2) the forwarding process of diffusion model with feature hashing, in Listing 2.

Listing 1: Pseudocode for GridBasedHashTable

```
1  # GridBasedHashTable Class Definition
2  Class GridBasedHashTable:
3      # Initializes the class with parameters for the hash table configuration
4      Constructor(delta_c: List, delta_t: Float, N: List, max_queue_length: Int,
            hash_table_size: Int):
5          # Spatial and temporal grid sizes and constants for hashing
6          Store delta_c, delta_t, and N as tensors
7          # Maximum queue length for each hash table entry and overall size
8          Store max_queue_length and hash_table_size
9          # Initialize hash table as a list of queues, one per hash table entry
10         hash_table ← list of deques, each with maxlen=max_queue_length
11
12     # Computes a raw hash index based on spatial-temporal key
13     def compute_hash_index_raw(key: Tensor) -> Int:
14         # Applies hashing formula to compute index based on key
15         i, j, k = floor(key[:3] / self.delta_c)
16         l = floor(key[3] / self.delta_t)
17         idx = i + self.N[0] * j + self.N[1] * k + self.N[2] * l
18         return idx
19
20     # Modulo operation to ensure index within hash table size
21     def compute_hash_index(key: Tensor) -> Int:
22         # Modulo hash_table_size to find actual index in hash table
23         idx = self.compute_hash_index_raw(key)
24         return idx % self.hash_table_size
25
26     # Appends feature data to the hash table, associated with spatial-temporal key and latent
27     def append(key: Tensor, feature: Tensor):
28         # Finds hash table index for given key
29         idx ← compute_hash_index(key)
30         # Appends the key, meta key, and feature as a tuple to the specified queue
31         hash_table[idx].append((key, feature))
32
33     # Queries the hash table for data matching a spatial-temporal key and meta key
34     def query(key: Tensor, meta_key: Tensor) -> Tensor or None:
35         # Finds hash table index for the query key
36         idx ← compute_hash_index(key)
37         # Retrieves the queue of data at the computed index
38         queue ← hash_table[idx]
39
40         # If the queue is empty, indicates no data for key
41         if queue is empty:
42             return None
43
44         # Extracts noisy latent and features from the queue for comparison
45         Unpack features from queue
46         # Computes distances between the query meta_key and stored meta_keys
47         Compute distances and apply softmax to derive weights
48         # Aggregates features based on weights to get a single output
49         Aggregate features using weights and return as aggregated output
```

Listing 2: Pseudocode for U-Net Inference with Feature Hashing

```
1  # function for U-Net forward pass with Feature Hashing (Example for Zero-123)
2  def forward_unet(x_in, vae_emb, t, t_in, cc_emb, polar, azimuth, radius, cache, cache_layer_id,
        cache_block_id):
3      Initialize prv_features to None
4      # Create a key tensor for caching based on stacking input parameters
5      keys ← [t[:batch_size], polar, azimuth, radius]
6
7      # Conditionally update cache based on a predefined probability
8      if random.random() < cache probability:
9          # Query the cache for each item in the batch
10         for each item k in keys:
11             prv_feature ← query hash table with key k
12
13             # Store retrieved hashed features
14             Update prv_features with hashed features
```

```
15
16    # Determine if new features need to be cached
17    append ← prv_features is None
18
19    # Perform U-Net prediction with potential use of cached features
20    (noise_pred, prv_features) ← unet(prv_features, other inputs...)
21
22    # Update cache with new features if necessary
23    if append:
24        for each item f in prv_features:
25            Cache new features f in the hash table
26
27    return noise_pred
```

## G  RELATED WORK

**3D Generation Model.** The development of 3D generative models has become a focal point in the computer vision. Typically, these models are trained to produce the parameters that define 3D representations. This approach has been successfully applied across several larger-scale models using extensive and diverse datasets for generating voxel representation Wu et al. (2016), point cloud Achlioptas et al. (2018); Nichol et al. (2022), implicit function Jun & Nichol (2023a), tri-plane Shue et al. (2023); Xu et al. (2024). Despite these advances, scalability continues to be a formidable challenge, primarily due to data volume and computational resource constraints. A promising solution to this issue lies in leveraging 2D generative models to enhance and optimize 3D representations. Recently, diffusion-based models, particularly those involving score distillation into 3D representations Poole et al. (2023), represent significant progress. However, these methods are often constrained by lengthy optimization processes.

**Efficient Diffusion Model.** Diffusion models, known for their iterative denoising process for image generation, are pivotal yet time-intensive. There has been a substantial body of work aimed at accelerating these models. This acceleration can be approached from two angles: firstly, by reducing the sampling steps through advanced sampling mechanisms Song et al. (2021); Bao et al. (2022); Liu et al. (2022); Lu et al. (2022) or timestep distillation Salimans & Ho (2022); Song et al. (2023), which decreases the number of required sampling steps. The second approach focuses on minimizing the computational demands of each model inference. This can be achieved by developing smaller diffusion models Kim et al. (2023); Yang et al. (2023); Fang et al. (2023) or reusing features from adjacent steps Ma et al. (2023); Li et al. (2023b), thereby enhancing efficiency without compromising effectiveness. However, the application of these techniques to 3D generative tasks remains largely unexplored.

**Hashing Techniques.** Hashing, pivotal in computational and storage efficiency, involves converting variable-sized inputs into fixed-size hash code via *hash functions*. These code index a *hash table*, enabling fast and consistent data access. Widely used in file systems, hashing has proven effective in a variety of applications, like 3D representation Nießner et al. (2013); Müller et al. (2022); Girish et al. (2023); Xie et al. (2023), neural network compression Chen et al. (2015); Kitaev et al. (2020), using hashing as a components in deep network Roller et al. (2021) and neural network-based hash function development Lai et al. (2015); Zhu et al. (2016); Cao et al. (2017); Li et al. (2017). Our study explores the application of hashing to retrieve features from 3D generation. By adopting this technique, we aim to reduce computational overhead for repeated diffusion sampling and speed up the creation of realistic 3D objects.

