# OpenReview forum: "Hash3D: Training-free Acceleration for 3D Generation"
_ICLR.cc/2025/Conference — ICLR 2025 Conference Withdrawn Submission_

### Official Review · Reviewer_JN1U · 2024-10-27

**Soundness:** 2
**Presentation:** 3
**Contribution:** 2
**Rating:** 5
**Confidence:** 2

**Summary:**

This paper tackles the challenge of 3D object generation using score distillation sampling (SDS) and proposes a method called Hash3D. This approach aims to speed up the optimization process by utilizing a hashing technique to reuse feature maps across nearby timesteps and camera angles. The experiments indicate that this method can accelerate optimization by 1.5 to 4 times without significantly compromising quality.

**Strengths:**

- The paper presents a plug-and-play, training-free acceleration method for diffusion-based text-to-3D and image-to-3D models. By leveraging the redundancy in diffusion models for nearby views and timesteps, it employs a hash table structure to enhance efficiency without sacrificing quality.
- Hash3D uses adaptive grid-based hashing for feature retrieval, which effectively reduces computational demands across different views and times.
- Experimental results demonstrate that Hash3D can speed up the generation process by 1.5 to 4 times while maintaining comparable quality.

**Weaknesses:**

While the paper introduces a new method aimed at speeding up SDS-based 3D object generation, several significant concerns arise:

- The feature similarity analysis relies on the Zero-123 model, which may not be applicable to other diffusion models. On page 4 (L204), the authors assert that similar patterns are observable in models like Stable Diffusion (Rombach et al., 2022). However, the methodology and visualizations used in this analysis are unclear.
- The data presented in Fig. 3 does not convincingly support the assumption that diffusion models exhibit redundant features for images generated from similar camera positions and timesteps. The blurred images produced at a delta angle of 5 degrees suggest that simply retrieving similar features may lead to a loss of surface detail in the generated objects.
- The need to maintain a memory hash table introduces several hyperparameters, and the overall speed-up is marginal (e.g., 1.5x on top of DreamFusion). Furthermore, the quality of the generated objects appears low (see Fig. 6), especially compared to state-of-the-art methods that offer faster runtimes.
- There are more effective strategies for improving speed, such as enhanced geometry initialization, efficient representations, and optimization techniques. This method does not seem to present a practical solution for accelerating 3D generation.

**Questions:**

Refer to weaknesses for details.
- Clarification of the feature similarity analysis is needed.
- The proposed method complicates the optimization process by introducing many hyperparameters, and the acceleration rate is very limited while the generation quality is low.
- The authors need to justify how the proposed method can preserve fine details in complex surfaces.

---

### Official Review · Reviewer_NsQs · 2024-11-01

**Soundness:** 3
**Presentation:** 2
**Contribution:** 2
**Rating:** 5
**Confidence:** 1

**Summary:**

This paper presents a training-free method, called Hash3D, to accelerate SDS-based methods. The authors begin by analyzing the hypothesis of feature redundancy in the denoising process of diffusion models across different timesteps and views of a 3D model. They provide an experiment to support this hypothesis. Since SDS relies on this process to generate guidance, the authors propose using grid hashing to store and reuse features when sampling similar timesteps and camera positions. To improve the robustness of this method for various objects, they introduce an adaptive approach to building the grid hash. Extensive results demonstrate the efficiency of this proposed method.

**Strengths:**

Here is the strengths of this paper:
- This paper introduces grid hashing to store diffusion features for acceleration, which is a novel idea to improve the efficiency of SDS-based methods. Although somewhat counterintuitive, the authors conduct extensive experiments to demonstrate its effectiveness.
- To enhance the impact of grid hashing, the paper proposes an adaptive method to adjust the grid size, further optimizing performance.
- The paper presents multiple experiments to evaluate the effectiveness of the proposed approach.

**Weaknesses:**

This paper presents a novel and general acceleration method for generation and evaluation in SDS-based models. However, I have several concerns from different perspectives:

**Experiments:**

ln 409:  distinct “train taillights” in row 4. WHERE IS ROW 4?

Ln 445: In the second row of Fig. 6, the Hash3D result significantly degrades generation quality (e.g., the rabbit's face and the consistency of the horse). This raises concerns about the robustness of this algorithm. A similar issue occurs in the "Batman" experiment with Fantasia3D. The authors should provide an explanation for this degradation.

This paper only compares the proposed algorithm using traditional SDS. Since it claims to be a universal acceleration for SDS-based methods, it should also be evaluated with several advanced SDS methods. Some examples for consideration include:

- Zhu, Joseph, and Peiye Zhuang. "Hifa: High-fidelity text-to-3D with advanced diffusion guidance." ICLR 2024.
- Wang, Zhengyi, et al. "ProlificDreamer: High-fidelity and diverse text-to-3D generation with variational score distillation." NeurIPS 2024.
- Liang, Yixun, et al. "LucidDreamer: Towards high-fidelity text-to-3D generation via interval score matching." CVPR 2024.

The authors should select at least one advanced SDS method for evaluation (it is no matter if it is the listed papers) to substantiate the claim of universal acceleration and demonstrate its effectiveness in various applications.

**Present:**

ln 212: The explanation in Synthesizing Novel Views for Free is somewhat confusing. For instance, "reusing and interpolating scores from precomputed nearby cameras" leaves questions about what "scores" refers to and how "interpolating" generates a novel view.  Also, the notation at angles (θ, φ) = (10 ± δ, 90) is how many angles? the notations make the "angles" seens like 10 with several offsets and 90. however, in the following sentense, "... synthesized a **third** view" so it seens like you only generate 2 views, one in 10 and other in 90? The authors should clarify this paragraph for better understanding.

**Typo:**
- ln 034: per se -> per scene ?

- ln 181: double comma

- ln 308: non-differentiale -> non-differentiable

- ln 326: HS -> Hash3D?

**Questions:**

Although this paper presents a relatively novel idea for accelerating SDS methods, it lacks experiments involving advanced SDS techniques, and some aspects of the presentation are unclear. Therefore, I am giving it a rating of 5 and encourage the authors to address the issues and concerns highlighted in the weaknesses section.

---

### Official Review · Reviewer_XaDG · 2024-11-03

**Soundness:** 2
**Presentation:** 3
**Contribution:** 2
**Rating:** 5
**Confidence:** 4

**Summary:**

This paper proposes a novel training-free acceleration module for optimization-based 3D generation tasks. In particular, the authors firstly analyse the calculation redundancy of the diffusion based optimization process, probing on which they propose to reuse similar previously-computed features to accelerate the current iteration, in a gird-based hashing manner. Furthermore, the authors utilize an adaptive hashing strategy to more efficiently retrieve features. The experimental results show the effectiveness of the proposed method on accelerating 3D generation.

**Strengths:**

* The method works as a plug-and-play module for various optimization-based 3D generation methods, without the requirement of further training.
* The proposed module can largely accelerate the process of 3D generation without significantly compromising the visual quality.
* The adaptive grid hashing technique effectively improves robustness of the proposed method when working on different cases.

**Weaknesses:**

* There exists inconsistency between the scenario which the analysises in section 3.1 are conducted on, and which this work actually addresses. The former is a purely 2D task while the latter is facing a 3D task with the targeted object evolving. Since the optimized 3D object keeps changing during the generation procedure (especially in the initial stage), could the hypotheses, temporal similarity and spatial similarity, still be held?
* The proposed method seems to introduce numerical error to the optimization process, because it uses (worse) prediction results from previous optimization steps to update the current step. It is evidenced by several noisy visual results (e.g., case 1,4 of SDS+3DGS group and case 2,4 of Fantasia3d group in Figure 6).
* The paper mentions the existence of other kinds of works for acceleration of 3D generation, but the experimental results lack comparison with them.
* While the results in Figure 3 have shown unaccepetable outcome when $\delta>5$, why the chosen window size candidates (\[10, 20, 30\]) are all larger than 5?

Reasons for current ratings:

While the submission creates a novel training-free acceleration module for optimization-based 3D generation tasks, the evaluations for the base hypotheses and comparison with baselines of this work are not conducted soundly enough. I think the authors should further polish this work in the above mentioned aspects.

**Questions:**

Besides the questions mentioned in Weaknesses, I have another question. When forwarding the clean input to large noise levels (e.g., t>700), the noisy latent should be dominated by the given noise since it has low SNR. It seems that the authors did not use same noise for each iteration, so I want to know how is the feature similarity of using different noise when t is large?

---

### Official Review · Reviewer_64j1 · 2024-11-04

**Soundness:** 2
**Presentation:** 2
**Contribution:** 2
**Rating:** 5
**Confidence:** 5

**Summary:**

Hash3D proposed a novel feature reusing strategy for score distillation sampling.
By hashing and reusing these feature maps across nearby timesteps and camera angles, Hash3D can speed up existing methods by $1.5 \sim 4 \times$.
Also, the feature reusing strategy can improve the visual quality of the generation.
The comprehensive experiments demonstrate the high-efficient acceleration.

**Strengths:**

1. The motivation is clear, and the results are convincing.
2. The adaptive grid size adjustment helps Hash3D not have to design the grid size for every generation.

**Weaknesses:**

1. The technical contribution is a little insufficient. Hash3D seems to be a simple extension of DeepCache[a].
2. The visual quality improvement is too marginal. The variance is already much more significant than the improvement claimed in the paper.
3. The acceleration rate is similar to DeepCache. It is hard to tell if the technical contributions in Hash3D really help compared with only caching the diffusion feature based on denoising steps, i.e., DeepCache.


Reference:
[a] Ma, Xinyin, Gongfan Fang, and Xinchao Wang. "Deepcache: Accelerating diffusion models for free." Proceedings of the IEEE/CVF Conference on Computer Vision and Pattern Recognition. 2024.

**Questions:**

1. typo in L089: $1.3 \to 1.5$ according to the abstract and conclusion in introduction.
2. The authors should provide the ablation `only discretize the diffusion timestep` vs. `only discretize the view direction` to ensure that the view direction hash table can contribute to the optimization acceleration.
3. Does Hash3D help the model converge faster? Since the authors claim they set the hash probability as $10$% for the feature query and we need to run the whole U-net when feature updating, the maximal acceleration should be $\sim \times 1.11$. Otherwise, the definition of hash probability in L265 is a typo.
4. The definition of noisy input $x$ is not clear. Also, the authors should provide the quantitative results of `w/o Content-based Aggregation`.

---

### Note · Authors · 2024-11-13

I have read and agree with the venue's withdrawal policy on behalf of myself and my co-authors.